# A Canadian Perspective on Systemic Therapy for Recurrent or Metastatic Nasopharyngeal Carcinoma

**DOI:** 10.3390/curroncol32010048

**Published:** 2025-01-17

**Authors:** Anna Spreafico, Eric Winquist, Cheryl Ho, Brian O’Sullivan, Nathaniel Bouganim, Neil Chua, Sarah Doucette, Lillian L. Siu, Desiree Hao

**Affiliations:** 1Division of Medical Oncology and Hematology, Princess Margaret Cancer Centre, University Health Network, Toronto, ON M5G 2M9, Canada; 2Division of Medical Oncology, Department of Oncology, Western University, London, ON N6A 5W9, Canada; 3Verspeeten Family Cancer Centre at London Health Sciences Centre, London, ON N6A 5W9, Canada; 4Department of Medical Oncology, BC Cancer, University of British Columbia, Vancouver, BC V5Z 4E6, Canada; 5Department of Radiology, Radio-Oncology and Nuclear Medicine, Centre Hospitalier de l’Université de Montréal (CHUM), Université de Montréal, Montreal, QC H2X 0C1, Canada; 6Department of Radiation Oncology, University of Toronto, Toronto, ON M5T 1P5, Canada; 7Radiation Medicine Program, Princess Margaret Cancer Centre, University Health Network, Toronto, ON M5G 2M9, Canada; 8Department of Oncology, McGill University Health Centre, Montreal, QC H3A 1G1, Canada; 9Department of Oncology, Cross Cancer Institute, University of Alberta, Edmonton, AB T6G 1Z2, Canada; 10Impact Medicom Inc., Toronto, ON M6S 3K2, Canada; 11Department of Medical Oncology, Arthur JE Child Comprehensive Centre, Cumming School of Medicine, University of Calgary, Calgary, AB T2N 5G2, Canada

**Keywords:** nasopharyngeal carcinoma, head and neck cancer, Epstein-Barr virus, human papillomavirus, immunotherapy

## Abstract

Although the majority of patients with nasopharyngeal carcinoma (NPC) present with early-stage or locoregional disease that can be treated with definitive radiotherapy, approximately 20% of patients experience disease recurrence, and 15% present with metastatic disease that is not amenable to curative therapy. Management of patients with recurrent or metastatic (r/m) NPC who are not candidates for local salvage therapy is challenging in Canada, as there is uncertainty in extrapolating evidence that is largely generated from Southeast China to non-endemic regions such as Canada. Currently, treatment options in Canada are limited to chemotherapy regimens that can only achieve short-term response and prolongation of survival. The addition of anti-PD-1 monoclonal antibodies to chemotherapy has been shown to extend progression-free survival in recurrent r/m NPC compared to chemotherapy alone; however, approval of PD-1 inhibitors in Canada has lagged behind other jurisdictions where NPC is non-endemic. This paper reviews the current systemic treatment landscape for r/m NPC in Canada, highlights unmet treatment needs for patients who are not candidates for curative therapy, and discusses the challenges and opportunities that lie in emerging therapies.

## 1. Introduction

Nasopharyngeal carcinoma (NPC) is a malignant tumor that arises from the epithelial lining of the nasopharynx. While NPC shares some characteristics with other head and neck cancers, it has distinct epidemiology, clinical behavior, and therapeutic strategies. The geographical distribution of NPC is highly variable, with markedly higher incidence in endemic regions such as North and East Africa and Southeast Asia, particularly Southern China [1]. In these regions, the annual incidence can range from 3–30 cases per 100,000, whereas non-endemic areas, including North America and Europe, generally report rates below 1 in 100,000 [1,2,3,4]. Canada is considered a non-endemic region for NPC occurrence; however, in certain populations, such as the Inuit populations in Nunavut and the Northwest Territories, an incidence of around 10 per 100,000 is reported [5].

The two most common histological subtypes in NPC are keratinizing squamous cell carcinoma and non-keratinizing carcinoma, with the latter being further divided into differentiated and undifferentiated subtypes. These subtypes are distributed differently across populations and regions, likely due to variations in genetic predispositions and environmental exposures. Notably, non-keratinizing NPC accounts for over 95% of cases in endemic areas, with most of these patients having undifferentiated histology [3]. For this reason, the majority of clinical trial data currently available focus on non-keratinizing NPC. In non-endemic regions, keratinizing NPC is relatively more common, constituting 20–50% of cases depending on the population studied, similar to the rates reported for undifferentiated non-keratinized NPC in these regions [3,4,6,7,8]. Differentiated non-keratinized NPC is also more common in non-endemic regions, accounting for approximately 20% of cases [4,8].

Risk factors for the development of NPC include genetic ancestry, consumption of salted fish, tobacco use (in well-differentiated NPC in low-risk populations), alcohol use, and Epstein-Barr Virus (EBV) exposure [9,10]. While undifferentiated non-keratinizing NPC tumors are almost exclusively positive for EBV, particularly in endemic areas, keratinizing NPC is rarely EBV positive, especially in Western countries [11]. The incidence of EBV-related differentiated non-keratinizing NPC is poorly characterized; however, its incidence appears to be rising in North America [8]. Human papillomavirus (HPV) has more recently been described as a risk factor for NPC, particularly in non-endemic areas, where high-risk HPV has been identified in 25–75% of EBV-negative cases [7,12,13,14]. In a retrospective analysis of a single tertiary institution cohort, Huang et al. suggested some apparent clinical differences between EBV- and HPV-related NPC, including a tendency for the latter to have larger primary tumors, symptomatic skull base involvement requiring pain management, and a slightly higher rate of distant metastasis [13]. Given the near mutual exclusivity of EBV- and HPV-related NPC, some North American centers routinely screen for high-risk HPV via p16 immunohistochemistry in EBV-negative NPC. However, beyond remaining alert to the noted clinical differences that could influence radiotherapy planning approaches [13], the implications of HPV-positivity on the management of NPC remain unclear.

Over 70% of NPC cases are diagnosed in early or locally advanced stages [15], where radiotherapy, with or without chemotherapy, can achieve long-term remission [16]. Approximately 20% of patients experience disease recurrence following curative intent therapy and up to 15% present with distant metastases at the time of diagnosis [6,17,18]. In cases of local recurrence, salvage surgery and/or re-irradiation should be considered based on resectability criteria and predicted risk/patient tolerance of late radiotherapy toxicity. Salvage surgery is often not possible when the tumor is close to the carotid artery or there is extensive skull base involvement, while re-irradiation is challenging if the tumor is adjacent to important dose-limiting surrounding structures that have likely been exposed to radiotherapy with initial treatment [19,20]. Systemic therapy is indicated for these individuals with r/m NPC who are not candidates for further curative local therapy; however, access to effective treatment options in Canada is currently limited and lags behind other similar jurisdictions. This paper aims to review the current Canadian treatment landscape for r/m NPC that is not amenable to curative therapy and to discuss the challenges and opportunities that lie in emerging therapies.

## 2. Chemotherapy and Radiation Therapy in the Palliative Setting

As described above, the benefits and risks of definitive salvage therapy for locally recurrent NPC should be carefully considered with the patient and multidisciplinary team, facilitated by the use of available international guidelines [19,20]. A recently published guideline supports endoscopic nasopharyngectomy as the preferred option for resectable locally recurrent NPC, based on data from a meta-analysis suggesting that it may improve survival compared to intensity-modulated radiation therapy (IMRT), with or without chemotherapy, in cases of recurrent (r)T3 to rT4 NPC (5-year overall survival: 52% vs. 31%) [19,21]. This guideline also suggests that post-operative re-irradiation following surgical salvage should be considered for patients with positive or close margins with optimal post-operative doses ranging around 60 Gy and using hyperfractionation in an attempt to further reduce toxicity. In patients not eligible for surgery, re-irradiation with or without chemotherapy can be considered [16].

For patients with r/m NPC who are not candidates for salvage surgery or re-irradiation, chemotherapy is the main treatment used in Canada in both the first-line and subsequent-line settings. Platinum-based chemotherapy with gemcitabine is the standard of care for first-line r/m NPC. This is supported by results from the phase III GEM20110714 study, conducted in China. In this study, six cycles of gemcitabine-cisplatin demonstrated significantly prolonged PFS (HR 0.55; 95% CI 0.44–0.68) and OS (HR 0.72; 95% CI, 0.58–0.90) compared with cisplatin-fluorouracil in patients with EBV-positive r/m NPC [22,23]. In cases where gemcitabine-cisplatin may not be suitable due to comorbidities and/or expected tolerability, carboplatin may be used instead of cisplatin; otherwise, cisplatin/carboplatin with fluorouracil, paclitaxel, or capecitabine are reasonable alternatives [24,25,26,27,28].

The benefit of administering gemcitabine as maintenance therapy is unclear but is used in some centers based on extrapolation from clinical trials evaluating other maintenance therapy regimens. This includes a phase 3 trial by Liu et al., which found that maintenance capecitabine led to significantly prolonged PFS compared to best supportive care in patients with newly diagnosed metastatic NPC who had achieved disease control after 4 to 6 cycles of paclitaxel, cisplatin, and capecitabine (median PFS 35.9 vs. 8.2 months; HR 0.44; 95% CI, 0.26–0.74) [29]. A randomized study from 5 centers in China also demonstrated a significantly improved PFS (16.9 vs. 9.3 months, *p* < 0.001) and OS (33.6 vs. 20.6 months, *p* < 0.001) with maintenance tegafur-gimeracil-oteracil (S1) compared with observation after a response to first-line platinum-based chemotherapy in patients with undifferentiated non-keratinizing r/m NPC [30]. Current international guidelines do not provide recommendations on maintenance therapy following platinum-gemcitabine in r/m NPC; however, given these data, the fact that NPC is highly chemosensitive, and that effective second-line therapies are limited, anecdotally, many clinicians will offer maintenance gemcitabine following platinum chemotherapy.

Radiation therapy as a palliative approach also has an evolving role in the management of r/m NPC following chemotherapy. Although not considered curative in this context, locoregional radiotherapy in doses similar to definitive-style radiation therapy can be used in de novo metastatic NPC to prolong survival [16]. In a phase 3 trial of 126 patients with de novo metastatic NPC who responded to the first 3 cycles of platinum-5-fluorouracil, the addition of intensity-modulated radiotherapy to the primary and nodal regions (50–70 Gy in 33 fractions, 5 times per week) after 6 cycles of chemotherapy resulted in improved OS (24-month OS: 76.4% vs. 54.5%) and PFS (12.4 vs. 6.7 months) compared with chemotherapy alone [31]. Radiation therapy for the primary tumor in patients with oligometastatic NPC who achieved some degree of tumor shrinkage with chemotherapy is also supported by retrospective studies [32,33,34,35]. Additional trials evaluating the benefit of radiotherapy following the current standard, gemcitabine-cisplatin, are needed.

Palliative radiation therapy to metastatic sites is typically used for symptom control [36]. Another strategy being explored is the delivery of stereotactic body radiation (SBRT) to oligometastases in combination with systemic treatment. To date, the only trial to report on this strategy is a small (n = 39) randomized phase 2 study exploring immunotherapy (camrelizumab) with or without metastasis-directed therapy (MDT) that included SBRT to at least one metastatic lesion [37]. The primary endpoint was to assess a potential abscopal effect through an improvement in the objective response rate (ORR) of unirradiated lesions in patients receiving immunotherapy who also received radiotherapy for other lesions. However, the study failed to meet the primary end-point, reporting an ORR of 26.3% in unirradiated lesions receiving MDT and 30.0% in the camrelizumab-alone group [37].

## 3. Immunotherapy with Anti-PD-1 Agents

Immune checkpoint blockade is an attractive strategy in EBV-associated NPC, given its favorable immune landscape, characterized by an abundance of tumor-infiltrating lymphocytes and high expression of PD-L1 and regulators of T-cell activation [38]. Single-agent anti-PD1 monoclonal antibodies have demonstrated overall response rates of 20–43% in previously treated r/m NPC [39,40,41,42,43] (Table 1). This is based on phase 2 trials of nivolumab and pembrolizumab, which included patients from non-endemic regions [39,40], and phase 1 or 2 trials of toripalimab, camrelizumab, and tislelizumab, which were conducted exclusively in China [41,43,44]. Pembrolizumab has been the only PD-1 inhibitor to be assessed in a randomized phase 3 trial (KEYNOTE-122) in patients with platinum-pretreated r/m NPC [45] (Table 2). In this trial, 14% of the enrolled population were from North America. Although a statistically significant improvement in PFS and OS compared with the investigator’s choice of capecitabine, gemcitabine, or docetaxel was not observed, pembrolizumab demonstrated a more favorable safety profile, with fewer grade 3–5 treatment-related adverse events (AEs) reported compared with the chemotherapy arm (10.3% vs. 43.8%).

Platinum-chemotherapy can increase the sensitivity of cancer cells to anti-PD-1 therapy through improved immunogenicity as a result of neoantigen release after tumor cell death, as well as chemotherapy-induced modulation of the tumor microenvironment and immune cells [46]. This supports the potential for an additive effect of anti-PD-1 therapy with gemcitabine-platinum in patients with r/m NPC. Three phase 3 randomized controlled trials have evaluated the PD-1 inhibitors toripalimab (JUPITER-02), camrelizumab (CAPTAIN-1st), and tislelizumab (RATIONALE-309) in combination with gemcitabine-cisplatin, followed by PD-1 inhibitor maintenance therapy, in patients with r/m NPC receiving first-line systemic therapy (Table 2) [47,48,49]. All studies took place in Asian centers, where most patients had a diagnosis of non-keratinizing NPC. All three studies met their primary endpoints by demonstrating improved PFS for the PD-1 inhibitor arms compared with gemcitabine-cisplatin and placebo maintenance. In subgroup analyses, a PFS benefit for PD-1 inhibitor therapy was observed regardless of baseline EBV levels or PD-L1 expression [47,48,49]. Although none of these studies were powered to detect a statistically significant difference in OS, an updated analysis of JUPITER-02 reported a reduction in risk of death for the toripalimab arm (HR 0.63 [95% CI 0.45–0.89]), while an updated analysis of RATIONALE-309 also found a reduction in risk of death for tislelizumab, although the confidence interval crossed 1 (HR 0.73 [95% CI 0.51–1.05]; Table 2) [47,50]. In a meta-analysis of these three phase 3 trials, the risk of death was significantly reduced with PD-1 inhibitors compared with chemotherapy alone (HR 0.63; 95% CI: 0.47–0.84) [51].

PD-1 inhibitors plus platinum-gemcitabine are generally well-tolerated, with similar rates of serious adverse events compared with chemotherapy alone, although, as expected, immune-related AEs occur at a higher frequency, most commonly hypothyroidism and rash [47,48,49]. Additional studies combining PD-1 inhibitors with other novel agents, chemotherapy combinations, and radiotherapy are ongoing, which will help to determine the optimal role of immunotherapy in the treatment of r/m NPC (Table 3).

**Table 1 curroncol-32-00048-t001:** Phase 1 or 2 trials of single-agent anti-PD-1 monoclonal antibodies in patients with previously treated recurrent or metastatic nasopharyngeal carcinoma.

Trial Name/Follow-Up	N	Arms	Race	Subtypes	ORR, %	Median PFS, Months	Median OS, Months
KEYNOTE-028 [39] Median follow-up: 20 months	27	Pembrolizumab	Asian: 63% White: 22% Black: 7% American Indian or Alaskan native: 4%	Keratinizing: 22% Non-keratinizing, undifferentiated: 30% Non-keratinizing, differentiated: 37% Other: 11%	25.9	3.7	16.5
NCI-9742 [40] Median follow-up: 12.5 months	44	Nivolumab	Asian: 82% African American: 2% Native Hawaiian/ Pacific Islander: 2% White: 9%	Non-keratinizing, undifferentiated: 56% Non-keratinizing, differentiated: 27% Other: 18%	20.5	2.8	17.1
NCT02721589 [41] Median follow-up: 9.9 months	93	Camrelizumab	Asian: 100%	Keratinizing: 16% Non-keratinizing, undifferentiated: 69% Non-keratinizing, differentiated: 13% Unclassified: 2%	34	5.6	Not reported
NCT04068519 [42,44] Median follow-up: 4.8 months	21	Tislelizumab	Asian: 100%	Non-keratinizing, undifferentiated: 76% Non-keratinizing, differentiated: 10% Unknown: 14%	43	10.4	Not estimable
POLARIS-02 [43] Median follow-up: Not reported	190	Toripalimab	Asian: 100%	Keratinizing: 4% Non-keratinizing: 96%	20.5	1.9	17.4

ORR, overall response rate; OS, overall survival; PFS, progression-free survival.

**Table 2 curroncol-32-00048-t002:** Phase 3 studies of immune checkpoint inhibitors in recurrent/metastatic nasopharyngeal carcinoma.

Study Name/Follow-Up	Arms	N	Region	Subtypes	ORR, %	Median PFS, Months	PFS E vs. C, HR (95% CI)	Median OS, Months	OS E vs. C, HR (95% CI)
First-line setting									
JUPITER-02 NCT03581786 [47] Median follow-up: 36.0 months	E: toripalimab + Gem-Cis→ toripalimab maintenance C: Placebo + Gem-Cis → placebo maintenance	E: 146 C: 143	Asia: 100%	Keratinizing: 1% Non-keratinizing: 99% Other: <1%	E: 78.8 C: 67.1	E: 21.4 C: 8.2	0.52 (0.37–0.73)	E: NR C: 33.7	0.63 (0.45–0.89)
CAPTAIN-1st NCT03707509 [48] Median follow-up: 15.6 months	E: camrelizumab + Gem-Cis → camrelizumab maintenance C: Placebo + Gem-Cis → placebo maintenance	E: 134 C: 129	Asia: 100%	Keratinizing: <1% Non-keratinizing: 99% (differentiated, 16%; undifferentiated, 82%) Other: <1%	E: 87.3 C: 80.6	E: 10.8 C: 6.9	0.51 (0.37–0.69)	E: NR C: 22.6	0.67 (0.41–1.11)
RATIONALE-309 NCT03924986 [49,50] Median follow-up: 41.4 months	E: tislelizumab + Gem-Cis → tislelizumab maintenance C: Placebo + Gem-Cis → placebo maintenance	E: 131 C: 132	Asia: 100%	Keratinizing: 7% Non-keratinizing: 89% (differentiated, 15%; undifferentiated, 74%) Unclassified: 5%	E: 69.5 C: 55.3	E: 9.6 C: 7.4	0.53 (0.39–0.71)	E: 45.3 C: 31.8	0.73 (0.51–1.05)
Platinum-pretreated in recurrent/metastatic setting									
KEYNOTE-122 NCT02611960 [45] Median follow-up: 45.1 months	E: pembrolizumab C: capecitabine, gemcitabine, or docetaxel	E: 117 C: 116	Asia: 86% North America: 14%	Non-keratinizing: 100% All EBV positive	E: 21.4 C: 23.3	E: 4.1 C: 5.5	1.28 (0.94–1.75)	E: 17.2 C: 15.3	0.90 (0.67–1.19)

C, control arm; CI, confidence interval; E, experimental arm; EBV, Epstein-Barr Virus; Gem-Cis, gemcitabine-cisplatin; HR, hazard ratio; NR, not reached; ORR, overall response rate; OS, overall survival; PFS, progression-free survival.

**Table 3 curroncol-32-00048-t003:** Active and planned clinical trials of interest in r/m NPC who are not amenable to curative treatment (as of 20 November 2024).

Trial	Phase	Treatment Arms	Target	Status	Region
First-line					
NCT05294172	3	Gem-Cis+ tagitanlimab vs. Gem-Cis + Placebo	PD-1	Active, not recruiting	China
NCT06383780 ^a^	3	Gem-Platinum + Tislelizumab vs. TPC + Tislelizumab	PD-1	Recruiting	China
NCT06177301	3	Tislelizumab + Gem-Cape vs. Tislelizumab + Gem-Cis	PD-1	Recruiting	China
NCT05869227	3	Maintenance: Toripalimab + Capecitabine vs. Toripalimab + Placebo	PD-1	Recruiting	China
NCT05854849	3	Camrelizumab + Gem + apatinib vs. Camrelizumab + Gem-Cis	PD-1/ VEGFR2	Recruiting	China
NCT04974398	3	Gem-Cis + penpulimab vs. Gem-Cis + Placebo	PD-1	Recruiting	International
NCT04890522	2/3	Triprilimab + Gem-Cis vs. Triprilimab + Cis-5-FU	PD-1	Not yet recruiting	China
NCT06486220	3	Cis-5-FU + anti-PD-1 mAb + FMT vs. Cis-5-FU + anti-PD-1 mAb	PD-1/gut microbiome	Not yet recruiting	China
NCT06669611	3	Camrelizumab + Gem-Platinum vs. Camrelizumab + TPC	PD-1	Not yet recruiting	China
NCT06457503 RTOG 3521 (TRANSPARENT)	4	Toripalimab + Gem-Platinum	PD-1	Recruiting	North America
NCT06029270 (REMAIN-NRG-HN011)	2	Maintenance after Gem-Cis + nivolumab: Nivolumab vs. Nivolumab + relatlimab	PD-1, LAG-3	Recruiting	International
NCT05385926 ^b^	2	Following response to 3–6 cycles of chemotherapy plus immunotherapy: Radiotherapy + anti-PD-1 maintenance	PD-1	Recruiting	China
Second/subsequent-line					
NCT06664983 ^c^	3	TPC + cadonilimab vs. TPC	PD-1/ CTLA-4	Active, not recruiting	China
NCT06241599	2/3	Gem-Platinum/PFLL + cadonilimab vs. Gem-Platinum/PFLL + placebo	PD-1/ CTLA-4	Recruiting	China
NCT06118333 ^c^	3	BL-B01D1 vs. Capecitabine/gemcitabine/docetaxel	EGFR–HER3	Recruiting	China
NCT05904080 ^c^	2	Cabozantinib + nivolumab + ipilimumab vs. Nivolumab + ipilimumab	PD-1/ CTLA-4/VEGFR2	Recruiting	United States
NCT06629597 ^c^	3	YL201 vs. Docetaxel/capecitabine/gemcitabine	B7-H3	Not yet recruiting	China

^a^ Patients must have bone metastases. ^b^ Patients with de novo metastatic disease. ^c^ Patients must have failed prior anti-PD-1 therapy. Cis-5-FU, cisplatin-5-fluorouracil; FMT, Fecal microbiota transplant; Gem, gemcitabine; Gem-Cape, gemcitabine-capecitabine; Gem-Cis, gemcitabine-cisplatin; PFLL, cisplatin, fluorouracil, and high-dose l-leucovorin; TPC, NAB-paclitaxel, platinum chemotherapy, capecitabine.

## 4. Targeted Therapies and Other Emerging Agents

Targeted therapies and other novel agents continue to be evaluated in r/m NPC. Given the increased expression of EGFR in approximately 80% of NPC cases [52], it has been explored as a therapeutic target but has limited use in current treatment paradigms. Cetuximab, an anti-EGFR monoclonal antibody, has demonstrated marginal activity in patients with pretreated r/m NPC, achieving an ORR of 11.7%, median PFS of 81 days, and median OS of 233 days in combination with carboplatin in a phase 2 study [53]. Other EGFR tyrosine kinase inhibitors such as gefitinib and erlotinib demonstrated minimal efficacy in this setting [54,55]. More recently, the EGFR-targeted antibody-drug conjugate (ADC) MRG003 has demonstrated an ORR of 55.2% in a phase 2a study of patients with r/m NPC whose disease had failed prior platinum and/or PD-1 therapy [56]. Preliminary results from another phase 2 study combining MRG003 with the PD-1 inhibitor pucotenlimab demonstrated an ORR of 77.8% in 9 patients with EGFR-positive r/m NPC who progressed after first-line treatment of PD-1 plus platinum-based chemotherapy [57]. After promising phase 1 data, the bispecific EGFR-HER3-targeted ADC, BL-B01D1, is currently being studied in a phase 3 trial, compared with chemotherapy, for patients with previously treated r/m NPC who progressed on PD-1 inhibitor therapy (NCT06118333, Table 3) [58].

VEGFR is another molecular target of interest in NPC, based on its frequent overexpression and role in angiogenesis and tumor metastasis. Initial studies with VEGFR inhibitors sorafenib, pazopanib, and sunitinib demonstrated modest efficacy but had a high incidence of hemorrhage, with fatal hemorrhagic events occurring [59,60,61]. Due to this safety risk, subsequent studies of VEGFR inhibitors axitinib and apatinib excluded patients with local recurrence or vascular invasion who are at higher risk of hemorrhage, resulting in lower incidences of this adverse event [62,63]. The use of apatinib in combination with camrelizumab achieved ORRs between 39–66% in phase 2 studies in heavily pretreated r/m NPC [64,65]. This regimen is currently being explored in a phase 3 study for the first-line treatment of r/m NPC (NCT05854849, Table 3). A randomized phase 2 study is also assessing nivolumab plus ipilimumab, with or without the VEGFR2 inhibitor cabozantinib, in NPC patients with disease progression after gemcitabine-cisplatin plus a PD-1 inhibitor (NCT05904080).

Another target of interest is the immunoregulatory protein B7-H3, which is upregulated on EBV-infected NPC cells compared to normal tissue [66]. In a phase 1 study, YL201, an ADC targeting B7-H3, achieved an ORR of 45.9% in 61 heavily pre-treated patients with metastatic NPC [67]. This agent is currently being studied in a phase 3 trial compared with chemotherapy for patients with previously treated r/m NPC who progressed on PD-1 inhibitor therapy (NCT06629597, Table 3). Other B7-H3 targeted agents are also being developed in solid tumors [68].

Several immunotherapy approaches which use T-cells to target EBV antigens on NPC cells are under investigation. Infusion of autologous EBV-specific cytotoxic T-cells in patients with NPC has demonstrated reasonable efficacy and minimal toxicity in small studies of patients with NPC [69,70]; however, in the VANCE trial, gemcitabine-carboplatin followed by EBV-cytotoxic T-cells did not demonstrate an OS or PFS benefit compared with gemcitabine-carboplatin alone [71]. Recently, tabelecleucel, an allogeneic, EBV-specific T-cell immunotherapy, has demonstrated promising activity in patients with r/m NPC, both alone and in combination with pembrolizumab, in phase 1 studies and is being actively explored in other EBV-positive malignancies [72,73].

Chimeric Antigen Receptor (CAR) T-cell therapies have also gained interest in a number of solid tumors and are being explored in r/m NPC, targeting both EBV-specific and other tumor antigens [24]. These include BRG01, an autologous EBV-targeting CAR T-cell therapy, which demonstrated promising efficacy and excellent tolerability in a phase 1 study in previously treated r/m NPC and is now the first CAR T-cell therapy to be cleared for investigation in a phase 2 trial in the U.S. and China [74,75].

Given the role of EBV in the development of NPC, vaccine therapy has been investigated in several clinical studies. Dendritic cell vaccines generated to express EBV targets and stimulate T-cell activity against EBV proteins have been investigated in phase I studies over the last two decades, with some clinical benefits observed [76,77,78,79]; however, larger randomized trials are needed. Recently, the FDA has approved the EBV-targeted mRNA vaccine WGc-043 for an investigational new drug application, based on investigator-initiated trials in NPC and natural killer T-cell lymphoma, which showed improved efficacy and safety for WGc-043 compared with other mRNA vaccine therapies [80].

## 5. Canadian Perspective

In Canada, the standard of care for r/m NPC in the first-line setting primarily involves chemotherapy with platinum-gemcitabine, although patient comorbidities, performance status, and patient goals and preferences may dictate alternative chemotherapy selection (Figure 1). As well, when gemcitabine-cisplatin is used as neoadjuvant treatment, it is unclear how effective this regimen will be for these patients if they have a recurrence, or how long the platinum-free interval should be before considering using it in the metastatic setting. At relapse, patients typically only have access to generic chemotherapy drugs. Enrollment in a clinical trial in this context would be preferred, but specific NPC trials are not frequently available in Canada.

The NCCN and ESMO-EURACAN guidelines recommend that PD-1 inhibitors, in combination with gemcitabine-cisplatin, be considered in first-line therapy for r/m NPC [36,81]. Currently, toripalimab, camrelizumab, and tislelizumab are all approved in combination with platinum-gemcitabine for first-line treatment and as monotherapy for subsequent-line treatment for patients with r/m NPC in China. Only toripalimab has been approved for both these indications in the United States, the European Union, and India [82,83,84]. This marks the first therapeutic advancement in NPC beyond chemotherapy in a non-endemic region. However, no PD-1 inhibitor is approved yet in Canada for r/m NPC, underscoring the gap between global standards and access to oncology drugs in Canada.

It is important to acknowledge the uncertainty in extrapolating results from the first-line, phase 3 PD-1 inhibitor trials, which were conducted exclusively in Asian centers, to the Canadian population. In general, optimizing treatment for Canadian NPC patients is challenging, given the rarity of the disease in North America; thus, there is limited regional representation in clinical trials. For instance, for the current standard of care, platinum-gemcitabine, a randomized phase 3 trial has only been conducted in an endemic region with EBV-positive, predominantly non-keratinizing NPC, although this regimen had also been evaluated in small retrospective and prospective studies from non-endemic regions [85,86].

The extent to which genetic and environmental factors, histological differences, and regional practices contribute to the differences in survival seen between endemic and non-endemic NPC populations requires further study. Non-endemic countries have a higher proportion of patients with keratinizing NPC, which is associated with poorer prognosis than non-keratinizing NPC [87]. This may contribute to the overall poorer prognosis for patients with NPC in non-endemic versus endemic regions. However, this is not necessarily the case in Canadian cities such as Vancouver or Toronto, which have a higher rate of Asian patients with non-keratinizing NPC (80–90%) and OS rates that more closely resemble those in Asian populations [6,85]. Conversely, in Inuit populations in Greenland and Denmark, OS rates are typically lower despite the majority of these patients having EBV-positive, non-keratinizing NPC, which may be extrapolated to the Inuit populations in Canada, where NPC is endemic [88,89]. Whether the benefit of adding PD-1 inhibitors to platinum-chemotherapy in patients with r/m NPC is similar in different histological subtypes and populations is unclear. The RATIONALE-309 trial studying tislelizumab enrolled more keratinizing NPC compared with other PD-1 inhibitor trials (7% vs. 1%, respectively), which is closer to the distribution of NPC subtypes in Canada, but subgroups were still too small for meaningful interpretation of efficacy by histology [49].

Another factor that may impact prognosis is HPV status. Association of NPC with high-risk HPV is more common in non-endemic regions, particularly in Caucasians. Several retrospective cohort studies have explored whether prognosis is different in HPV-positive versus EBV-positive NPC, with some studies reporting no difference in OS and others reporting shorter survival for HPV-positive patients [14,90].

Canadian clinicians may consider treating HPV-positive nasopharyngeal tumors with similar approaches to those used for HPV-positive head and neck squamous cell carcinoma (HNSCC), particularly when the tumor is located near the oropharynx [13]. However, evidence to support this approach remains limited and is not discussed in international guidelines. It will be important to continue to tease out the impact of intrinsic and extrinsic factors on survival and how these factors may impact response to different therapies. Exploration of treatment approaches for non-endemic NPC, including keratinizing, as well as HPV-positive NPC, would be helpful to confirm the benefit of PD-1 inhibitors in a broader range of Canadian NPC cases. This may be addressed in the upcoming North American, single-arm TRANSPARENT (RTOG 3521) study in which patients will receive cisplatin-gemcitabine plus toripalimab in first-line r/m NPC (Table 3).

## 6. Future Directions for Optimizing PD-1 Inhibitor Use in r/m NPC

Several gaps in knowledge remain that are important for clarifying the role of PD-1 inhibitors in r/m NPC, regardless of population. Firstly, elucidating biomarkers to predict response to PD-1 would be helpful to better select which patients may benefit most from therapy. The impact of PD-L1 expression on clinical outcomes for patients with NPC who are receiving PD-1 inhibitors in combination with platinum-chemotherapy was assessed in the JUPITER-02 and RATIONALE-309 studies, with the former defining PD-L1 positivity as the presence of membrane staining of any intensity in 1% or greater of tumor cells or immune cells and the latter evaluating tumor positive score at a cut-off of 10% (high/low expression) [47,49]. In this context, neither study found the PFS benefit for PD-1 inhibitors to be significantly different between PD-L1 expression subgroups. Additionally, in the KEYNOTE-122 study, PD-L1 expression, using a combined positive score (including tumor and immune cells) at cut-offs of 1%, 10%, and 20%, did not appear to correlate with a PFS advantage for pembrolizumab monotherapy in patients with platinum-pretreated r/m NPC [45]. Other immune markers have been suggested, which need further validation in prospective studies [49].

Determination of the optimal length of PD-1 inhibitor maintenance is also needed, and whether this could be guided by biomarkers should be explored. In the CAPTAIN-1st trial, early clearance of plasma EBV was associated with improved PFS in a post hoc analysis; thus, it would be interesting to further explore this as a marker to guide treatment escalation/de-escalation in patients who are EBV-positive [48]. Identifying the patients most likely to respond to PD-1 inhibitor-based therapy and optimizing treatment duration are particularly important questions to answer, given that PD-1 inhibitors are associated with adverse events that can diminish quality of life and the high cost of these drugs would place a financial burden on publicly funded healthcare systems such as those across Canada.

The use of PD-1 inhibitors for the first-line treatment of r/m NPC would create new questions about how to treat patients after progression and identify the mechanisms of resistance to PD-1 inhibition and how they can be overcome. Comparisons between PD-1 maintenance and gemcitabine maintenance therapy are also needed, along with insights on the relative efficacy of PD-1 inhibitor maintenance versus, or in addition to, definitive-style palliative radiation therapy. As patients with recurrent NPC progressing within 6 months of curative chemoradiation therapy were excluded from the first-line trials, the role of PD-1 inhibitors in combination with platinum-gemcitabine in this population is uncertain, particularly as it is unclear whether these patients would remain sensitive to platinum-chemotherapy. The appropriate interval and combination regimen for PD-1 inhibitors for patients with NPC recurring following definitive therapy requires further study.

## 7. Conclusions

The prognosis for patients with r/m NPC remains sub-optimal in both endemic and non-endemic regions. As a result, there is a high unmet need for additional therapeutic options. There are significant disparities in access to immunotherapy as part of first-line treatment for r/m NPC, with Canada lagging behind the U.S., the European Union, and China. Advocacy to streamline the Canadian approval process for novel therapeutics in NPC is important, and regulatory bodies may consider adopting a higher degree of flexibility when reviewing new therapies in NPC, given that it is considered a rare disease in Canada, which limits opportunities to generate Canadian-specific data [91]. Continued and cooperative research efforts are needed to address the knowledge gaps in non-endemic NPC, including exploration of different biological and environmental factors impacting survival and response to therapy that may help guide the management of r/m NPC in Canada.

## Figures and Tables

**Figure 1 curroncol-32-00048-f001:**
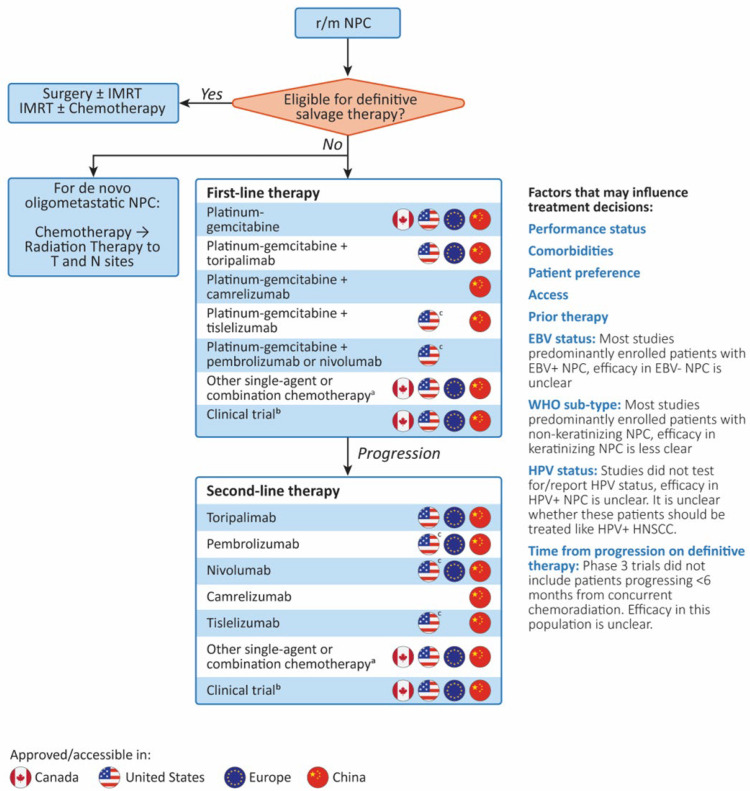
Treatment algorithm for r/m NPC and treatment access in Canada, the United States, Europe, and China. ^a^ Includes single agents/combinations of paclitaxel, docetaxel, 5-FU, capecitabine, gemcitabine, irinotecan, vinorelbine, cisplatin, oxaliplatin, carboplatin, ifosfamide, doxorubicin, methotrexate, and/or cetuximab, depending on region and preference. ^b^ Access to clinical trials specifically for patients with NPC is more limited in Canada, the United States, and Europe. ^c^ Listed as a recommended regimen in the National Comprehensive Cancer Network guideline. Not approved by the FDA for this specific indication. EBV, Epstein-Barr virus; HNSCC, head and neck squamous cell carcinoma; HPV, human papillomavirus; IMRT, intensity-modulated radiation therapy; NPC, nasopharyngeal carcinoma; r/m, recurrent/metastatic; WHO, World Health Organization.

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
