# Peer review of "A Canadian Perspective on Systemic Therapy for Recurrent or Metastatic Nasopharyngeal Carcinoma"

_curroncol, 2025, doi:10.3390/curroncol32010048_

Round 1

Reviewer 1 Report

Comments and Suggestions for Authors

This paper reviews the current systemic treatment landscape for recurrent/metastatic nasopharyngeal carcinoma (r/m NPC) in Canada, highlights unmet treatment needs for patients who are not candidates for curative therapy, and discusses the challenges and opportunities associated with emerging therapies. Undoubtedly, this review provides valuable guidance for future research and clinical practice regarding r/m NPC in Canada. Several key points should be noted as follows:

1) In the 'Introduction' section, it would be advantageous to enhance the overall background information regarding NPC. A recent publication offers an innovative perspective that conceptualizes NPC as an ecological disease—a multidimensional spatiotemporal pathological ecosystem (https://pubmed.ncbi.nlm.nih.gov/37056571/). It is advisable to review this ESI-high cited paper for its comprehensive and novel insights into NPC.

2) How about the side effects associated with all systemic therapies, including targeted therapies and other emerging agents. Notably, the prognosis for recurrent/metastatic nasopharyngeal carcinoma (r/m NPC) remains suboptimal, particularly when considering the significant impact on patients' quality of life and the substantial financial burden.

3) Have we ever contemplated such a question: Perhaps for such patients in the relevant sections, if no treatment is administered, there might be no change in their survival time, or it could even be better. In other words, in reality, some of these current treatment methods actually decelerate the progression of the patients' diseases. It is quite clear that a considerable number of patients, even through immunotherapy, will exhibit no response. why does such a situation occur among these non-responding patients ?

4) What was the ultimate cause of death for these patients, regardless of the treatment they received? Was it the cancer itself or the treatment itself?

5) Since ethnic groups vary and the common pathological tissue types are also different, what do you think will be the differences in the systemic treatments to be adopted in the future compared with the prevalent regions?  

Author Response

Response to Reviewer 1

Comment 1
In the 'Introduction' section, it would be advantageous to enhance the overall background information regarding NPC. A recent publication offers an innovative perspective that conceptualizes NPC as an ecological disease—a multidimensional spatiotemporal pathological ecosystem (https://pubmed.ncbi.nlm.nih.gov/37056571/). It is advisable to review this ESI-high cited paper for its comprehensive and novel insights into NPC.

Response to Comment 1
The authors have reviewed the suggested paper and feel that it should not be included for the following reasons. As our current manuscript focuses on systemic treatment for relapsed/refractory NPC, we feel it is out of scope to provide additional details on the pathophysiology of NPC (which we feel is the focus of the suggested paper). This is particularly true given the current length of our introduction. Additionally, the recommended paper only provided some suggestions on how understanding the pathophysiology of NPC may guide the development of novel therapeutics, which is out of scope for our current paper that describes treatments that are available and are being investigated in clinical trial.

Comment 2
How about the side effects associated with all systemic therapies, including targeted therapies and other emerging agents. Notably, the prognosis for recurrent/metastatic nasopharyngeal carcinoma (r/m NPC) remains suboptimal, particularly when considering the significant impact on patients' quality of life and the substantial financial burden.

Response to comment 2
Although the current manuscript focuses on the discussion of efficacy of systemic therapies, some mention of safety impacts or tolerability can be found throughout the manuscript (see lines: 119-121; 171-175; 196-199; 235-238 ). To address the safety and financial implications of PD-1 inhibitors, we have included the following sentence in section 6 (lines 373-378):  

“Identifying the patients most likely to respond to PD-1 inhibitor-based therapy and optimizing treatment duration are particularly important questions to answer given that PD-1 inhibitors are associated with adverse events that can diminish quality of life and the high cost of these drugs would place a financial burden on publicly funded healthcare systems like those across Canada.”

Comment 3
Have we ever contemplated such a question: Perhaps for such patients in the relevant sections, if no treatment is administered, there might be no change in their survival time, or it could even be better. In other words, in reality, some of these current treatment methods actually decelerate the progression of the patients' diseases. It is quite clear that a considerable number of patients, even through immunotherapy, will exhibit no response. why does such a situation occur among these non-responding patients ?

Response to comment 3
Although it is an interesting question, it is ultimately difficult to assess on an individual patient level whether that patient may or may not benefit from a treatment such as immunotherapy in terms of overall survival. In section 6 we acknowledge that biomarkers are needed to better understand which patients are likely to respond to PD-1 inhibitors, as currently assessed biomarkers have not been found to be predictive of response. We have included additional details on biomarker studies in lines 356-367:

“The impact of PD-L1 expression on clinical outcomes for patients with NPC who are receiving PD-1 inhibitors in combination with platinum-chemotherapy was assessed in the JUPITER-02 and RATIONALE-309 studies, with the former defining PD-L1 positivity as the presence of membrane staining of any intensity in 1% or greater of tumor cells or immune cells and the latter evaluating tumor positive score at a cut-off of 10% (high/low expression) [47, 49]. In this context, neither study found the PFS benefit for PD-1 inhibitors to be significantly different between PD-L1 expression subgroups. Additionally, in the KEYNOTE-122 study, PD-L1 expression, using a combined positive score (including tumour and immune cells) at cut-offs of 1%, 10%, and 20%, did not appear to correlate with a PFS advantage for pembrolizumab monotherapy in patients with platinum-pretreated r/m NPC [45]. Other immune markers have been suggested which need further validation in prospective studies [49].”  

Comment 4
What was the ultimate cause of death for these patients, regardless of the treatment they received? Was it the cancer itself or the treatment itself?

Response to comment 4
We do not feel that addressing the cause of death for patients in each individual study described in this review fits within the scope of our paper, particularly as not all publications have these details available or report whether death is considered related to a specific component of treatment. We do not feel that there is a significant signal for treatment-related death outside of what is already established for PD-1 inhibitors across multiple cancers that necessitates a discussion of this outcome in the current paper (typically <1% as per systemic review and meta-analysis by Wang et al, doi: 10.1001/jamaoncol.2019.0393)

Comment 5
Since ethnic groups vary and the common pathological tissue types are also different, what do you think will be the differences in the systemic treatments to be adopted in the future compared with the prevalent regions?

Response to comment 5
We feel this is an important question to answer and suggested in the original manuscript that future trials may help to address this question (e.g. lines: 342-352). This included a suggestion that for patients with HPV-related NPC in non-endemic regions, they may perhaps be best treated in a similar way to HPV-positive head and neck squamous cell carcinoma (HNSCC); however, we don’t have enough evidence to support this.

Reviewer 2 Report

Comments and Suggestions for Authors

Overall, the paper is good. It will not get attention from other clinicians and scientists outside Canada because this is just a Canadian problem. The authors have done a good job in summarising the issues for NPC in non-endemic regions.

Just a minor edit needed in the conclusion. No need to refer to Fig. 1. Just say what you want us to see.

Author Response

Response to Reviewer 2

Comment 1
Just a minor edit needed in the conclusion. No need to refer to Fig. 1. Just say what you want us to see.

Response to comment 1
We appreciate this comment and have removed the specific reference to figure 1 in the conclusion (line 401).

Reviewer 3 Report

Comments and Suggestions for Authors

Very little to say: this review is well organized, well written, and it comes from one of the leading cancer centers in the world. The tables and the figure are very informative, updated, and well organized. The English language is obviously perfect.

Just discuss futher two points:

- the issue of costs of these new immune-modulating drugs in the context of the many different healthcare systems you are referring to in Figure 1

- the role of PD1 scores such as the CPS in predicting the response to the newly available inhibitors, this remains an understudied issue

Author Response

Response to Reviewer 3

Comment 1
Discuss the issue of costs of these new immune-modulating drugs in the context of the many different healthcare systems you are referring to in Figure 1

Response to comment 1
Based on this comment we have included a sentence to emphasize the importance of optimizing patient selection and treatment duration for PD-1 inhibitors in NPC, particularly given the safety risks associated with these drugs and their high costs. As this paper is focussed on the Canadian perspective we mention the financial burden associated with high drug costs in the context of a publicly funded healthcare system. See lines: 373-378.    

“Identifying the patients most likely to respond to PD-1 inhibitor-based therapy and optimizing treatment duration are particularly important questions to answer given that PD-1 inhibitors are associated with adverse events that can diminish quality of life and the high cost of these drugs would place a financial burden on publicly funded healthcare systems like those across Canada.”

Comment 2
Discuss the role of PD1 scores such as the CPS in predicting the response to the newly available inhibitors, this remains an understudied issue

Response to comment 2
We had briefly mentioned the unreliability of PD-L1 expression in predicting the benefit of response to PD-1 inhibitors in NPC in section 6, but have now expanded this section to provide more details on which studies and which PD-L1 scoring systems were evaluated. See lines 357-367:

“Firstly, elucidating biomarkers to predict response to PD-1 would be helpful to better select which patients may benefit most from therapy. The impact of PD-L1 expression on clinical outcomes for patients with NPC who are receiving PD-1 inhibitors in combination with platinum-chemotherapy was assessed in the JUPITER-02 and RATIONALE-309 studies, with the former defining PD-L1 positivity as the presence of membrane staining of any intensity in 1% or greater of tumor cells or immune cells and the latter evaluating tumor positive score at a cut-off of 10% (high/low expression) [47, 49]. In this context, neither study found the PFS benefit for PD-1 inhibitors to be significantly different between PD-L1 expression subgroups. Additionally, in the KEYNOTE-122 study, PD-L1 expression, using a combined positive score (including tumour and immune cells) at cut-offs of 1%, 10%, and 20%, did not appear to correlate with a PFS advantage for pembrolizumab monotherapy in patients with platinum-pretreated r/m NPC [45]. Other immune markers have been suggested which need further validation in prospective studies [49].”

Round 2

Reviewer 1 Report

Comments and Suggestions for Authors

No other questions.